# Mitochondria Fusion upon SERCA Inhibition Prevents Activation of the NLRP3 Inflammasome in Human Monocytes

**DOI:** 10.3390/cells11030433

**Published:** 2022-01-27

**Authors:** Ana Catarina Pereira, Nuno Madeira, Sofia Morais, António Macedo, Maria Teresa Cruz, Cláudia M. F. Pereira

**Affiliations:** 1CNC—Center for Neuroscience and Cell Biology, CIBB—Center for Innovative Biomedicine and Biotechnology, University of Coimbra, Rua Larga—Faculdade de Medicina, 1°andar—Polo I Universidade de Coimbra, 3004-504 Coimbra, Portugal; anacatarinajpp@gmail.com (A.C.P.); trosete@ff.uc.pt (M.T.C.); 2Faculty of Medicine, University of Coimbra, 3000-548 Coimbra, Portugal; nunogmadeira@gmail.com (N.M.); sofiamorais86@gmail.com (S.M.); amacedo@ci.uc.pt (A.M.); 3CACC—Clinical Academic Center of Coimbra, 3004-561 Coimbra, Portugal; 4CIBIT—Coimbra Institute for Biomedical Imaging and Translational Research, University of Coimbra, Azinhaga de Santa Comba, 3000-548 Coimbra, Portugal; 5Department of Psychiatry, CHUC-UC—Centro Hospitalar e Universitário de Coimbra, 3004-561 Coimbra, Portugal; 6Faculty of Pharmacy, University of Coimbra, 3000-548 Coimbra, Portugal

**Keywords:** calcium homeostasis, endoplasmic reticulum (ER) stress, immune system, sterile inflammation, mitochondria dynamics

## Abstract

Sarco/endoplasmic reticulum Ca^2+^ ATPase (SERCA) is a crucial component of the cellular machinery responsible for Ca^2+^ homeostasis. The selective inhibition of SERCA by thapsigargin (TG) leads to perturbations in Ca^2+^ signaling, which can trigger endoplasmic reticulum (ER) stress. The unfolded protein response (UPR) pathway is activated in response to ER stress and induces an adaptive response to preserve cell survival or committee cells to programmed death, depending on stress duration and/or level. Early stages of ER stress stimulate mitochondrial metabolism to preserve survival but under chronic ER stress conditions, mitochondrial dysfunction is induced, which, in turn, can enhance inflammation through NLRP3 inflammasome activation. This study was aimed at investigating the role of SERCA inhibition on NLRP3 inflammasome activation in human monocytes, which was evaluated in primary monocytes isolated from healthy individuals and in the THP-1 human monocytic cell line. Findings obtained in both THP-1 and primary monocytes demonstrate that SERCA inhibition triggered by TG does not activate the NLRP3 inflammasome in these innate immune cells since IL-1β secretion was not affected. Results from THP-1 monocytes showing that SERCA inhibition increases mitochondrial Ca^2+^ content and fusion, in the absence of changes in ROS levels and membrane potential, support the view that human monocytes counteract ER stress that arises from inhibition of SERCA through modulation of mitochondrial morphology towards mitochondria fusion, thus preventing NLRP3 inflammasome activation. Overall, this work contributes to a better understanding of the molecular mechanisms that modulate the activity of the NLRP3 inflammasome leading to sterile inflammation, which are still poorly understood.

## 1. Introduction

Endoplasmic reticulum (ER) is the main intracellular Ca^2+^ store and the major site for protein synthesis, folding and maturation in eukaryotic cells [1]. The sarco/endoplasmic reticulum Ca^2+^ ATPase (SERCA) is a pump that belongs to the P-type ATPase family, which uses the energy produced by ATP hydrolysis for the active transport of Ca^2+^ ions from the cytosol into the sarcoplasmic reticulum or endoplasmic reticulum (ER) lumen, in the case of muscle or non-muscle cells, respectively. Based on its molecular mechanism of action, SERCA maintains a low cytosolic Ca^2+^ concentration playing a pivotal role in the regulation of Ca^2+^ homeostasis, which is crucial for cell signaling and survival [2,3]. Indeed, Ca^2+^ is a ubiquitous intracellular messenger able to coordinate various cellular functions; however, at high concentrations, Ca^2+^ becomes cytotoxic [4,5]. Inhibition of SERCA activity has been closely associated with the depletion of Ca^2+^ levels in ER reservoirs and, subsequently, the disruption of ER functions, namely the activity of the Ca^2+^-dependent ER chaperons, which leads to the accumulation of unfolded proteins in the ER lumen, triggering ER stress [5,6,7]. Thapsigargin (TG) is a specific SERCA inhibitor described as a known ER stressor and therefore as an activator of the unfolded protein response (UPR) cascade, which consists of a set of transcriptional and translational events triggered in response to cellular stress [8,9].

Regarding ER stress signaling, BiP/GRP78 chaperone, which is critical for ER protein quality control, is associated with the three ER stress sensors of the UPR: protein kinase R-like ER kinase (PERK), inositol-requiring enzyme 1 alpha (IRE1α), and transcription factor 6 (ATF6), keeping them inactive during resting conditions. Upon ER stress, BiP/GRP78 chaperone binds to unfolded proteins accumulated in the ER lumen, which leads to dissociation of BiP/GRP78 protein from the ER stress sensors, culminating in UPR activation [10,11]. Depending on the level of cell stress, UPR orchestrates an adaptive response able to restore ER homeostasis and avoid cell death, or can trigger UPR-mediated apoptosis [9,12,13].

In addition to the UPR, cells also have other signaling mechanisms that act as strategic responses to help them deal with stressful conditions. The ER transmembrane chaperone Sigma-1 receptor (Sigma-1R) seems to be fundamental to trigger and fit anti-stress responses, since it can interact with multiple proteins and exhibit pleotropic effects [14,15]. For instance, Sigma-1R interacts with BiP/GRP78 chaperone, forming a complex at MAM surface. Upon ER Ca^2+^ depletion, which can arise from chronic ER stress, Sigma-1R dissociates from BiP/GRP78 in order to regulate Ca^2+^ flux from the ER into mitochondria via IP3 receptors (IP3Rs). In addition, Sigma-1R also influences bioenergetics and cellular survival through modulation of ER Ca^2+^ signaling, [16,17]. Recently, Sigma-1R was found particularly enriched in specialized ER-subdomains, called mitochondria-associated membranes (MAMs), which are contact sites resulting from the physical interaction of ER and mitochondria, thus suggesting its role in regulating ER-mitochondria communication [16].

Chronic/severe ER stress often triggers inflammatory responses, culminating in systemic metabolic dysfunction [18]. Over the last few years, multiprotein complexes called NLRP3 inflammasomes have been identified as central players in innate immune responses [19]. They are composed by a cytosolic pattern-recognition receptor, the Nod-like receptor pyrin domain containing 3 (NLRP3), the enzyme caspase-1, and the adaptor protein termed apoptosis-associated speck-like protein containing a CARD (ASC). The self-association of all these components leads to inflammasome activation, which is characterized by caspase-1-dependent maturation of the proinflammatory cytokines interleukin-1β (IL-1β) and interleukin-18 (IL-18). NLRP3 is the best-characterized inflammasome and has been defined as a driver of inflammation in response to mitochondrial damage. Classic NLRP3 activating stimuli include reactive oxygen species (ROS), release of mitochondrial DNA (mtDNA), cardiolipin externalization, release of cathepsins into the cytosol after lysosomal destabilization, and alterations in Ca^2+^ homeostasis [20,21,22]. Mitochondrial dysfunction is intrinsically linked to loss of mitochondrial membrane potential and disrupted mitochondrial network dynamics [23]. In healthy cells, mitochondrial morphology depends on a dynamic equilibrium between fission and fusion events. Mitochondrial fusion is coordinated by mitofusins (MFN1, MFN2), whereas dynamin-related protein 1 (DRP1) is defined as the major mediator of mitochondrial fission [24]. In addition to dynamic regulation of mitochondrial morphology, MFN2 also modulates ER-mitochondria tethering [25]. Induction of stress conditions has been closely associated with a dynamic remodeling of mitochondrial morphology [26,27]. Under acute ER stress conditions, a protective mechanism mediated by enhanced mitochondrial fusion can be stimulated to prevent pathologic mitochondrial fragmentation and promote mitochondrial metabolism, which is dependent on membrane potential [27]. Under severe stress conditions, there is an imbalance in the highly conserved dynamic processes towards mitochondrial fission, which is linked to the loss of mitochondrial membrane potential that acts as a signal to facilitate fragmentation and degradation of damaged mitochondria by mitophagy [26,28]. Recent studies suggest that perturbations in mitochondrial fission contribute to NLRP3 inflammasome-mediated inflammation [29,30].

This study aimed to investigate whether SERCA inhibition, and subsequent induction of ER stress, activates the NLRP3 inflammasome leading to sterile inflammation in human monocytes. Furthermore, the relationship between mitochondrial dysfunction and NLRP3 inflammasome activation under these conditions was also evaluated in human THP-1 monocytes.

## 2. Materials and Methods

### 2.1. Materials

The THP-1 human monocytic cell line (ATCC TIB-202) was bought from InvivoGen (Toulouse, France). RPMI 1640 medium, penicillin, streptomycin, thapsigargin, bicinchoninic acid (BCA) protein assay kit, and tetramethyl-rhodamine ethyl ester (TMRE) probe were obtained from Sigma Chemical Co. (St. Louis, MO, USA). Fetal bovine serum (FBS) was from Gibco, Life Technologies (Paisley, UK). The fluorescent probes MitoSOX red mitochondrial superoxide indicator, Hoechst 33342, and Rhodamine-2 acetoxymethyl ester (Rhod-2 AM) were obtained from Invitrogen (Eugene, OR, USA). Carbonyl cyanide p-trifluoromethoxyphenylhydrazone (FCCP) and the selective inhibitor RU 360 were obtained from Merck (Kenilworth, NJ, USA). All reagents used for culture primary monocytes including RPMI 1640 medium HEPES no glutamine, penicillin, streptomycin, sodium pyruvate, glutamax, non-essential amino acids, and heat-inactivated fetal bovine serum (FBS) were obtained from Gibco, Thermo Fisher Scientific (Waltham, MA, USA). Human monocytes isolation kit II was obtained from Miltenyi Biotec (Gladbach, Germany). Legend MAX Human IL-1β ELISA kit with precoated plates was obtained from Biolegend (San Diego, CA, USA). Protease- and phosphatase-inhibitor cocktails were obtained from Roche (Mannheim, Germany), and the NZY colour protein marker II was from NZYTech (Lisbon, Portugal). The alkaline phosphatase-linked secondary antibodies and the enhanced chemifluorescence (ECF) reagent were obtained from GE Healthcare (Chalfont St. Giles, UK), and the polyvinylidene difluoride membranes were from Millipore Corporation (Bedford, MA, USA). Antibodies against ATF4, CHOP, and Sigma-1R were from Cell Signaling Technology (Danvers, MA, USA). The anti-XBP1s and anti-DRP1 (Ser616) antibodies were from Biolegend (San Diego, CA, USA). Anti-Mitofusin 2 antibody was from Santa Cruz Biotechnology (Dallas, TX, USA), and anti-GRP78 was from BD Biosciences (San Jose, CA, USA). All other reagents were from Sigma Chemical Co. (St. Louis, MO, USA).

### 2.2. Methods

#### 2.2.1. THP-1 Cell Culture

The THP-1 human monocytic cell line was cultured and maintained in 75 cm^2^ flasks at a cell density between 0.5–1.0 × 10^6^ cells/mL in RPMI 1640 medium supplemented with 10% (*v*/*v*) heat-inactivated fetal bovine serum (FBS), 25 mM glucose, 10 mM Hepes, 1 mM sodium pyruvate, 100 U/mL penicillin, and 100 μg/mL streptomycin. Cells were maintained at 37 °C in a humidified incubator under an atmosphere containing 5% CO_2_. Cells were sub-cultured every 2–3 days and kept in culture for a maximum of 2 months.

#### 2.2.2. Isolation and Culture of Primary Human Monocytes

Peripheral blood (~20 mL) was collected by vein puncture from healthy male subjects, aged between 18 and 35 years old, upon written informed consent and study approval by the Ethical Committee of the Coimbra University Hospital (150/CES, 3 July).

Collected peripheral blood was diluted 2× by addition of sterile phosphate-buffered saline (PBS), pH 7.4, and then 20 mL of diluted blood was carefully added to 12 mL of ficoll-paque plus. After centrifugation at 1100× *g* for 20 min at room temperature (RT), with the centrifuge brake turned off, the mononuclear fraction was collected and subsequently diluted in 50 mL sterile PBS. A subsequent centrifugation at 300× *g* for 10 min at RT was used to pellet cells that were resuspended in 50 mL sterile PBS, and then they were centrifuged again at 200× *g* for 10 min at RT in order to remove platelets. Peripheral blood mononuclear cells (PBMCs) were then resuspended in 10 mL sterile PBS, and the number of cells were determined using an Eve Automatic Cell Counter (NanoEnTeK, Waltham, MA, USA).

Upon resuspension in MACS buffer (PBS, pH 7.2, supplemented with 0.5% (*v*/*v*) FBS and 2 mM EDTA), monocytes were then isolated by using a human monocyte isolation kit II according to the manufacturer’s instructions. Briefly, the mononuclear fraction was incubated with CD14 microbeads during 15 min at 4 °C. Pelleted cells were resuspended in MACS buffer and then centrifuged at 300× *g* for 10 min at RT. After resuspension again in MACS buffer, cell suspension was loaded onto a LS magnetic column placed in the magnetic field of a MACS separator and rinsed three times with the same buffer. Finally, the CD14^+^ fraction was eluted in MACS buffer, centrifuged at 300× *g* for 10 min at RT, and then cultured at a density of 0.8 × 10^6^ cells/mL in glutamine-free RPMI 1640 Medium, supplemented with HEPES, 10% (*v*/*v*) heat-inactivated FBS, 100 U/mL penicillin, 100 μg/mL streptomycin, 2 mM glutamax, 1 mM sodium pyruvate, and 0.1 mM non-essentials amino acids. Cells were maintained at 37 °C in a humidified incubator under an atmosphere containing 5% CO_2_.

#### 2.2.3. Total Cell Lysates Preparation

THP-1 monocytes were plated in a 6-well plate at a density of 2.4 × 10^6^ cells/well and treated in the absence or presence of 5 or 10 μM thapsigargin (TG) for the indicated time periods (1–24 h). After incubation, cells were washed with cold PBS, pH 7.4, and lysed on ice with ice-cold lysis RIPA buffer (50 mM Tris–HCl (pH 8.0), 1% (*v*/*v*) Nonidet P-40, 150 mM NaCl, 0.5% (*w*/*v*) sodium deoxycholate, 0.1% (*w*/*v*) SDS, 2 mM EDTA, and 1 mM DTT) freshly supplemented with a protease and phosphatase inhibitor cocktail. Nuclei and insoluble cell debris were removed by centrifugation at 12,000× *g* for 10 min at 4 °C, and the supernatant was collected and stored at −80 °C until further use.

The total protein amount was determined using the bicinchoninic acid (BCA) method. Then, proteins on cell lysates were denatured in sample buffer (5% (*w*/*v*) SDS, 0.125 M Tris-HCl pH 6.8, 20% (*v*/*v*) glycerol, 10% (*v*/*v*) 2-mercaptoethanol and bromophenol blue) by heating for 5 min at 95 °C.

#### 2.2.4. Western Blotting

Briefly, proteins (40 μg) were separated by electrophoresis on 10% (*v*/*v*) sodium dodecyl sulphate-polyacrylamide gels (SDS-PAGE) at 130 V for 60–75 min and then transferred to a methanol-activated polyvinylidene difluoride (PVDF) membrane by electroblotting using a Trans-Blot Cell wet transfer system (Bio-Rad, Hercules, CA, USA) at 400 mA for 3 h at 4 °C. After blocking with 5% (*w*/*v*) nonfat dry milk in Tris-buffered saline ((TBS): 150 mM NaCl, 25 mM Tris-HCl pH 7.6) containing 0.1% (*v*/*v*) Tween 20 (TBS-T) for 1 h, at RT, membranes were incubated with the primary antibodies diluted in TBS-T with 1% (*w*/*v*) nonfat dry milk overnight at 4 °C. After washing with TBS-T, membranes were incubated for 1 h at RT with alkaline phosphatase-conjugated anti-rabbit or anti-mouse (1:20,000) antibodies. After an additional washing step with TBS-T, the immune complexes were detected with the enhanced chemifluorescence (ECF) reagent using the scanning system Typhoon FLA 9000 (GE Healthcare, Chalfont St. Giles, UK). Total Lab TL 120 software (GE Healthcare, Chicago, IL, USA) was used to quantify the optical density of the bands. The results obtained were normalized to β-tubulin I protein loading control. At the end, results were normalized to control (untreated cells).

#### 2.2.5. Determination of Secreted IL-1β Levels in the Cell Supernatant

THP-1 human monocytic cell line was plated in 12-well plates at a density of 1.2 × 10^6^ cells/well and then treated with LPS for 24 h, prior to treatment with 5 or 10 μM TG for 8 h. As a positive control, THP-1 cells were exposed to 1 μg/mL LPS for 24 h plus 5 mM ATP for 30 min. Human primary monocytes were plated in a 96-well plate at a density of 0.12 × 10^6^ cells/well and then treated with 10 μM TG for 8 h in the presence or absence of LPS pre-incubated during 24 h. Briefly, untreated and treated cells were centrifuged at 400× *g* for 5 min, and the supernatants were collected and stored at −80 °C for subsequent use. IL-1β secretion in THP-1 monocytes and in primary human monocytes was measured with an ELISA kit according to the manufacturer’s instructions. Absorbance values were measured in a standard Synergy HT Multi Detection Microplate Reader (BioTek Instruments, Winooski, VT, USA) set to 450 nm and 570 nm wavelengths. IL-1β secreted levels were expressed as pg/mL.

#### 2.2.6. Determination of Mitochondrial Membrane Potential Assay

The fluorescent probe tetramethylrhodamine ethyl ester (TMRE) was used to detect alterations in mitochondrial membrane potential induced by TG in monocytes. For that, THP-1 cells were plated in a 48-well plate at a density of 0.3 × 10^6^ cells/well and then treated with 5 or 10 μM TG for 4 or 8 h. Cells treated with 50 μM carbonyl cyanide p-trifluoromethoxyphenylhydrazone (FCCP) for 10 min were used as a positive control for loss of mitochondrial membrane potential. After incubation for 30 min at 37 °C with 1 μM TMRE in PBS containing 0.2% (*w*/*v*) BSA, control and TG-treated cells were resuspended in PBS containing 0.2% (*w*/*v*) BSA, and the fluorescence was monitored with a Synergy HT Multi Detection Microplate Reader (BioTek Instruments, Winooski, VT, USA) set to 549 nm excitation and 575 nm emission wavelengths. Results were normalized to cells in the absence of TG exposure.

#### 2.2.7. Detection of Mitochondrial ROS Production

The fluorescent MitoSOX probe was used to evaluate the mitochondrial production of superoxide in control and TG-treated monocytes. For that, human THP-1 cells were plated in 12-well plates at a density of 1.2 × 10^6^ cells/well and then incubated in the absence or presence of 5 or 10 μM TG for 4 or 8 h. Cells were washed with PBS and subsequently incubated in Hanks’ Balanced Salt Solution (HBSS) containing 5 μM MitoSOX and 0.5 μg/mL Hoechst 33342 for 10 min at 37 °C. Once washed again with PBS, cells were resuspended in HBSS free of supplements. Finally, the fluorescent signal was monitored using an Observer Z.1 fluorescence microscope (Zeiss, Oberkochen, Germany). Images were captured using the Apochromat 63×/1.40 Oil DIC M27 objective. The results were normalized to cells in the absence of TG exposure.

#### 2.2.8. Transmission Electron Microscopy (TEM)

Electron microscopy was used to evaluate mitochondria morphological alterations in TG-treated human THP-1 monocytes. For this purpose, cells were plated in 6-well plates at a density of 2.4 × 10^6^ cells/well in a final volume of 3 mL and were then treated with 10 μM TG for 8 h. Briefly, cells were collected and centrifuged at 1008× *g* for 5 min. The supernatant was discarded, and pelleted cells were fixed with 2.5% (*w*/*v*) glutaraldehyde in 0.1 M sodium cacodylate buffer (pH 7.2) for 2 h. Cells were rinsed in the same buffer, and post-fixation was performed using 1% (*w*/*v*) osmium tetroxide for 1 h. After rinsing with buffer, samples were dehydrated in a graded ethanol series (70–100%). Following embedding in 2% (*w*/*v*) molten agar, samples were re-dehydrated in ethanol (30–100%), impregnated, and included in Epoxy resin (Fluka Analytical, Munich, Germany). Ultrathin sections were mounted on copper grids, and observations were carried out at 100 kV on Tecnai G^2^ Spirit BioTwin electron microscope (FEI, Hillsboro, OR, USA).

#### 2.2.9. Detection of Mitochondrial Calcium Content

The rhodamine-2 acetoxymethyl ester (Rhod-2/AM) fluorescent probe was used to determine the mitochondrial calcium content in controls and in monocytes upon TG treatment. For that purpose, human THP-1 cells were cultured in 12-well plates at a density of 1.2 × 10^6^ cells/well and treated in the absence or in the presence of 5 or 10 μM TG for 4 or 8 h. Cells treated with the selective inhibitor of mitochondrial calcium uniporter (MCU), RU 360 (10 μM, 1 h), were used as a positive control to Rhod-2 localization in the mitochondria.

After washing with PBS, cells were resuspended in HBSS containing 10 μM Rhod-2 AM and subsequently incubated for 45 min at 37 °C. Once washed again with PBS, cells were resuspended in HBSS free of supplements, and the fluorescence values were monitored with a standard spectrophotometer Synergy HT Multi Detection Microplate Reader (BioTek Instruments, Winooski, VT, USA) set to 552 nm excitation and 581 nm emission wavelengths. The results were normalized to cells in the absence of TG exposure.

#### 2.2.10. Evaluation of Cell Viability

Susceptibility towards TG-induced ER stress was evaluated in human monocytes by the resazurin assay. THP-1 cells were cultured in a 96-well plate at a density of 0.2 × 10^6^ cells/well and treated in the presence or absence of 5 or 10 μM TG for 24 h. Cells were incubated with resazurin solution (50 μM) 4 h before the end of treatment, at 37 °C. Absorbance values were measured at 570 and 600 nm in a Synergy HT Multi Detection Microplate Reader (BioTek Instruments, Winooski, VT, USA), and the final values were obtained from the subtraction of results determined at 600 nm from those measured at 570 nm. Cell viability was determined as percentage (%) of controls in the absence of TG.

#### 2.2.11. Statistical Analysis

Results are presented as mean ± standard error of the mean (SEM). Comparisons between two groups were made using Student’s unpaired *t* test with one-tailed *p* value. One-way ANOVA with unpaired Dunnett’s post hoc test was used for multiple comparisons. In both cases, a value of *p* < 0.05 was considered statistically significant. Statistical analysis was performed with Prism 7.0 (GraphPad Software, San Diego, CA, USA).

## 3. Results

### 3.1. SERCA Inhibition Induces ER Stress in Human Monocytes

First, induction of ER stress was evaluated in human monocytes treated with TG, a classic SERCA inhibitor. For that, activation of the ER stress-induced UPR signaling pathway was assessed in THP-1 cells treated with 5 or 10 μM TG in a time-dependent manner (1–24 h) by measuring the protein levels of UPR markers namely ATF4, XBP1s, GRP78, and CHOP. ATF4 protein levels were found significantly increased at all time points in TG-treated THP-1 monocytes when compared with controls cells (Figure 1A). A time-dependent increase in XBP1s levels was observed in TG-treated cells, which reached statistical significance at 24 h (Figure 1B). The levels of CHOP and GRP78 showed a tendency to increase, namely at 8 and 24 h of TG exposure, respectively, although it was not statistically significant (Figure 1C,D). The results showed that SERCA inhibition triggered by TG induces ER stress in human monocytes, as demonstrated by the activation of the UPR pathway.

### 3.2. SERCA Inhibition Is Not Able to Upregulate ER Stress Response Strategies

Given that Sigma-1R plays a key role in cellular stress signaling, namely under ER stress conditions, its protein levels were evaluated by WB in THP-1 monocytes treated with 5 or 10 μM TG in a time-dependent manner (1–24 h). No differences were observed in Sigma-1R levels between controls and TG-treated cells, indicating that ER stress induced by SERCA inhibition does not upregulate Sigma-1R in order to mitigate ER stress (Figure 2).

### 3.3. SERCA Inhibition Fails to Promote IL-1β Secretion in Human Monocytes

Afterwards, the role of SERCA inhibition on NLRP3 inflammasome activation was investigated by measuring the levels of IL-1β secreted by control and TG-treated THP-1 monocytes. In cells treated with LPS, the secretion of IL-1β increased by about 400-fold in comparison with control cells. The canonical NLRP3 activators LPS plus ATP augmented IL-1β secretion by about 200-fold in comparison with LPS-treated monocytes (Figure 3). No increment in IL-1β levels was detected in supernatants from THP-1 cells treated with LPS plus vehicle or treated with LPS plus TG, suggesting that TG is not able to activate the NLRP3 inflammasome that promotes IL-1β release to the extracellular space.

After demonstrating that TG-induced SERCA inhibition is not able to activate NLRP3 inflammasome in the THP-1 cell line of human monocytes, its effect was also investigated in primary monocytes isolated from peripheral blood (Figure 4). Differences were not observed when comparing the IL-1β levels secreted by control and TG-treated primary monocytes (Figure 4A). Given that NLRP3 inflammasome activation is classically described as a two-signal model, its activity was further assessed in LPS-primed cells. Once again, the secreted levels of IL-1β were similar between monocytes incubated with LPS alone or with LPS plus 10 μM TG (Figure 4B).

These findings indicate that TG-induced SERCA inhibition does not work as an activator of the NLRP3 inflammasome in primary monocytes as well, supporting that this effect is specific of this type of cell from the innate immune system.

### 3.4. NLRP3 Inflammasome-Associated Triggers Are Not Promoted by SERCA Inhibition

Once it was demonstrated that SERCA inhibition does not activate the NLRP3 inflammasome in human monocytes, some key mitochondrial events that have been described as triggers of inflammasome activation were evaluated, including ROS production, membrane depolarization, and fission. The alterations induced by TG on mitochondrial membrane potential were assessed with the TMRE fluorescent probe using FCCP as a positive control (Figure 5A). In cells treated with this uncoupler of the electron transport chain FCCP, TMRE fluorescence was approximately 50% of that displayed by control cells. On the other hand, no differences were observed in TMRE fluorescence between control and TG-treated cells at 4 and 8 h, suggesting that SERCA does not induce loss of mitochondrial membrane potential in THP-1 monocytes.

The mitochondrial ROS production in human monocytes was assessed through the MitoSOX probe (Figure 5B). Both control and TG-treated THP-1 cells exhibited similar levels of MitoSOX fluorescence, showing that SERCA inhibition induced by TG does not stimulate the accumulation of oxidant species within mitochondria.

Given that DRP1-mediated mitochondrial fission can trigger NLRP3 inflammasome activation, the levels of DPR1 phosphorylated at Ser616 were evaluated by WB in the absence or presence of SERCA inhibition in human monocytes. No relevant differences were observed in p-DPR1 levels between control and TG-treated THP-1 cells from 1–24 h of TG exposure (Figure 5C).

Taken together, these findings suggest that molecular triggers of NLRP3 inflammasome activation, such as mitochondrial ROS, mitochondrial membrane loss, and fission are not affected by SERCA inhibition in human monocytes.

### 3.5. SERCA Inhibition Increases Mitochondrial Fusion and Ca^2+^ Content

Then, the effect of SERCA inhibition on mitochondrial fusion was analyzed in human THP-1 monocytes. For that purpose, protein levels of Mitofusin 2, a protein mediator of mitochondria fusion were assessed by WB in controls and in TG-treated cells (Figure 6A). A significant upregulation of Mitofusin 2 was detected upon 8 and 24 h of exposure to TG when compared to control conditions, suggesting that TG promotes mitochondrial fusion. These findings were further supported by results obtained by TEM, showing that the presence of fused mitochondria is increased in monocytes treated with 10 µM TG when compared to untreated cells (Figure 6B).

Mitochondrial Ca^2+^ content was also evaluated under SERCA inhibition conditions in human monocytes, using the Rhod-2AM fluorescent probe (Figure 6C). When compared with control cells, a significant augment in Rhod-2AM fluorescence was found in THP-1 monocytes treated with TG, particularly after 4 and 8 h incubation, suggesting that SERCA inhibition in these innate immune cells is followed by a rise in mitochondrial Ca^2+^ content. Moreover, control monocytes were incubated with the selective inhibitor of the MCU, RU 360, in order to ensure that Rhod-2 fluorescence is specifically associated with mitochondria Ca^2+^ and is not result of probe mislocalization. A significant reduction in Rhod-2 fluorescence was observed in cells treated with RU 360 in comparison with untreated cells, demonstrating that mitochondria Ca^2+^ is the main source of the observed Rhod-2 fluorescence.

Finally, susceptibility of human monocytes towards SERCA inhibition was assessed by the resazurin assay in cells treated with 5 or 10 μM TG for 24 h. Both TG concentrations were shown to decrease cell viability by approximately 20% (Figure 6D).

## 4. Discussion

Activation of the NLRP3 inflammasome has been implicated in a wide range of diseases. However, the underlying mechanisms remain largely uncertain [31,32]. In order to clarify the signaling pathways that regulate the NLRP3 inflammasome in human monocytes, the role of SERCA inhibition induced by exposure to TG was evaluated in these cells from the innate immune system. For that purpose, both human THP-1 monocytic cell line and primary monocytes were used. The low yield of monocytes isolated from peripheral blood strongly impacted the number of parameters analyzed on these cells. To overcome this limitation, THP-1 human monocytes were used as an in vitro model, allowing us to further explore NLRP3 inflammasome activation.

First, it was demonstrated that inhibition of SERCA activates the ER stress-induced UPR in THP-1 monocytes. More specifically, TG activated both the PERK and IRE1α branches of the ER UPR, as shown by the upregulation of ATF4 and XBP1s, respectively, and ER stress was found to be an early event. The PERK and IRE1α UPR branches were the particular focus of this study, given their crucial role in triggering inflammatory responses during ER stress. Under stressful conditions, components of the PERK and IRE1α UPR arms interact with inflammatory signaling cascades, which results in the upregulation of inflammatory genes and subsequent induction of inflammatory responses. Activated IRE1 interacts with IKK and JNK, leading to activation of the inflammatory mediators NF-κB and AP-1. When activated by phosphorylation, eIF2α, which is a signaling mediator at the PERK UPR branch, is also able to activate NF-κB by inhibiting the translation of IκBα [33].

UPR activation in TG-treated cells was further supported by the tendency of increase in other ER stress markers such as GRP78 and CHOP. These data are in accordance with studies that have emerged over the years reporting consistently the activation of the UPR in TG-treated mammalian cells [8,34,35]. ATF4 transcription factor regulates the expression of numerous genes, including the pro-apoptotic factor CHOP. These mediators of the PERK UPR arm play an important role in committing cells to apoptosis under chronic ER stress conditions [35]. Our findings suggest that human monocytes are able to trigger strategies to cope with chronic ER stress upon SERCA inhibition, as indicated by the 17–25% decrease in viability observed in cells treated with 5 or 10 μM TG for 24 h, respectively. Based on ISO 10993-5:2009, “Biological evaluation of medical devices-Part5: Tests for in vitro cytotoxicity”, a reduction in cell viability below 30% is not considered a cytotoxic effect. Therefore, under the present experimental conditions, TG is non-cytotoxic.

Concerning the molecular mechanisms that could be involved in the adaptive response triggered by innate immune cells under stressful conditions, the focus was on Sigma-1R-mediated stress response. Structurally, Sigma-1R is an ER membrane protein that contains an *N*-terminus that faces the cytosol and a C-terminus that faces the ER lumen [36]. Its major action site is the ER, where it can bind specific signaling molecules, such as the ER chaperone GRP78/BiP [16,17,37]. Functionally, in its dormant state, Sigma-1R forms a complex with GRP78/BiP. Under ER stress conditions, Sigma-1R dissociates from GRP78 and regulates the chaperone activity at the ER by stabilizing the IRE1 and IP_3_Rs [16,38,39]. Mori and colleagues showed that Sigma-1R is able to associate with activated IRE1, which upregulates ER chaperones [38,39]. Regarding its action on IP_3_Rs, Hayashi and Su found that Sigma-1R-IP_3_R complexes prolong ER-to-mitochondria Ca^2+^ flux under chronic ER stress conditions. Moreover, it was observed that ER stress induced either by TG or tunicamycin, specific ER stressors, promotes the transient expression of Sigma-1R, which was not detected after 3 h. According to these authors, the Sigma-1R upregulation represents a cellular response to stress, and its inhibition 3 h upon ER stress induction can be explained by a decline in the stability of Sigma-1R protein and/or mRNA [16]. Our data suggest that SERCA inhibition does not activate Sigma-1R-mediated stress response in THP-1 monocytes.

TG is a specific blocker of the SERCA pump, which is crucial for reallocate released Ca^2+^ back into the ER and maintain Ca^2+^ homeostasis. Since Ca^2+^ can activate physiological and pathological signaling pathways, a tight regulation of intracellular Ca^2+^ concentrations is crucial to maintain proper Ca^2+^ dynamics and thus preserve normal cell function [4,5]. Due to its mechanism of action, TG administration has been intimately associated with a rapid depletion of ER Ca^2+^ stores and, subsequently, with an increase in intracellular Ca^2+^ levels [2,8]. Our findings showing enhanced levels of the fusion protein Mfn2 in TG-treated THP-1 cells, together with TEM representative images that show an increase in fused mitochondria in these cells, suggest that mitochondria fusion might be an adaptive response triggered by human monocytes to cope with stress induced by SERCA inhibition. The imbalance of mitochondrial dynamics towards mitochondrial fusion can be responsible for enhanced mitochondrial Ca^2+^ uptake observed in TG-treated THP-1 cells, which is required to activate enzymes involved in the TCA cycle and ATP production [40,41]. This hypothesis is supported by a recent study developed by Lebeau and collaborators that shows that TG-induced ER stress remodels mitochondrial morphology to prevent mitochondrial fragmentation and thus promote mitochondrial metabolism in response to ER stress. Moreover, it was demonstrated that the mechanism that protects mitochondria during ER stress is orchestrated through the PERK branch of the ER UPR. Upon activation, PERK selectively activates eIF2α by phosphorylation, which, in turn, promotes protective stress-induced mitochondrial hyperfusion [27]. This protective mechanism promotes elongation of mitochondria and avoids fragmentation that triggers mitophagy, and it was previously described as a cellular response to other types of stress, such as starvation, ribosome inhibition, and UV irradiation [42,43,44]. According to Sabouny et al., mitochondria hyperfusion mediated by the Keap1–Nrf2 pathway in response to stress involves the degradation of the mitochondrial fission protein DRP1 [44]. The enhanced mitochondrial fusion as a protective response to ER stress was further strengthened by data showing that mitochondrial morphology regulates mitochondrial Ca^2+^ uptake in order to preserve intracellular Ca^2+^ homeostasis [45]. These authors observed that the increment in mitochondrial fusion, which was achieved through the expression of an abnormal form of the fission protein DRP1, increases both mitochondrial Ca^2+^ uptake and Ca^2+^ retention capacity in C2C12 cells. Interestingly, they also reported a significant reduction in Ca^2+^ uptake and retention within mitochondria in cells expressing a fission phenotype that was induced by knockdown of the gene encoding the fusion protein Mfn2 [45]. Additionally, we observed that modulation of mitochondrial dynamics towards increased fusion is independent of alterations in mitochondrial membrane potential. This finding is also corroborated by Kowaltowski and coworkers, showing that the increase in mitochondria fusion in cells expressing abnormal DRP1 is independent of mitochondrial depolarization [45]. The preservation of mitochondrial membrane potential after TG treatment in THP-1 monocytes could be explained by the activity of Ca^2+^ extrusion systems that avoid mitochondrial Ca^2+^ overload [46].

ER stress affects mitochondrial function, and, depending on the level of stress, mitochondrial dysfunction can be promoted. Under chronic/severe ER stress, the extensive mitochondria Ca^2+^ influx is followed by mitochondrial fragmentation, increased ROS production, and loss of membrane potential, which can act as a signal to programmed cell death [47]. Together, the findings herein presented, showing absence of increased mitochondrial fission, membrane depolarization, or accumulation of ROS, indicate that SERCA inhibition in human monocytes does not promote mitochondrial damage, supporting that these innate immune cells are able to cope with stress. Furthermore, the above-mentioned mitochondrial alterations have been implicated in NLRP3-mediated inflammation [23]. Accumulation of mitochondrial ROS and disrupted mitochondrial membrane potential are classical activators of the NLRP3 inflammasome [20,21,22,23]. Although the role of mitochondrial dynamics in NLRP3 inflammasome activation is less explored, it has been proposed that fission of the mitochondrial network also plays a role [29]; however, the mechanisms involved are still controversial. On the one hand, Li and coworkers found that inhibition of DRP1-mediated mitochondrial fission blocks NOX2 oxidase signaling, preventing NLRP3 inflammasome activation in palmitate-treated endothelial cells [48]. NOX2 is a NADPH oxidase responsible for the production of ROS in many cells [49]. On the other hand, Park and colleagues demonstrated that activation of the NLRP3 inflammasome in response to LPS plus ATP is regulated by DRP1 in mouse bone marrow-derived macrophages. By silencing DRP1, these authors provided evidence that defective mitochondrial fission activates the NLRP3-dependent caspase-1 and subsequent IL-1β secretion [29]. Although there were no changes in relevant mitochondrial triggers of NLRP3 inflammasome activation upon SERCA inhibition in THP-1 human monocytes, secreted IL-1β levels were measured as a readout of NLRP3 inflammasome activation by TG-treated THP-1 cells. Under these conditions, extracellular IL-1β secretion was not promoted, and therefore, it can be concluded that NLRP3 inflammasome is not activated upon SERCA inhibition in THP-1 human monocytes. These findings obtained in THP-1 monocytes are supported by a previous study showing that IL-1β secretion in response to TG treatment is not detected in PMA-differentiated THP-1 macrophages [19]. Accordingly, the impact of SERCA inhibition on NLRP3 inflammasome was also evaluated in primary monocytes isolated from peripheral blood of healthy subjects exposed to TG, in the presence or absence of LPS priming. In both cases, extracellular IL-1β levels were not increased, supporting the view that absence of NLRP3 inflammasome activation is a specific early stress response of human monocytes triggered upon SERCA inhibition. Menu and co-workers have demonstrated that TG induces NLRP3 activation in murine LPS-primed bone marrow-derived macrophages [19]. Although both monocytes and macrophages are key components of the innate immune system, the differential response to stress induced by SERCA inhibition suggests that NLRP3 activation in the immune system occurs in a cell type-specific manner. The difference in the response between macrophages and monocytes may be explained by its functions on the innate immune system. As described in the literature, macrophages are the main innate immune cells involved in acute inflammatory responses, which are not so exacerbated in their precursors, the monocytes [50].

## 5. Conclusions

This work provides novel evidence that NLRP3 inflammasome is not activated by SERCA inhibition in human monocytes, either in primary monocytes or in the THP-1 monocytic cell line. Our findings from THP-1 cells showing increased mitochondrial fusion under these stressful conditions suggest that human monocytes trigger adaptive strategies through remodeling of mitochondrial morphology towards fusion in order to preserve metabolism and energy production. This mechanism, called mitochondrial hyperfusion, prevents mitochondrial dysfunction in response to ER stress caused by SERCA inhibition, namely ROS accumulation, membrane depolarization, and fission, thus avoiding NLRP3 inflammasome activation and, subsequently, the release of the pro-inflammatory cytokine IL-1β. Therefore, human monocytes can counteract early ER stress due to SERCA inhibition by modulating mitochondrial morphology and function in order to preserve energy production and cell survival and to prevent sterile inflammation induced by NLRP3 inflammasome activation.

## Figures and Tables

**Figure 1 cells-11-00433-f001:**
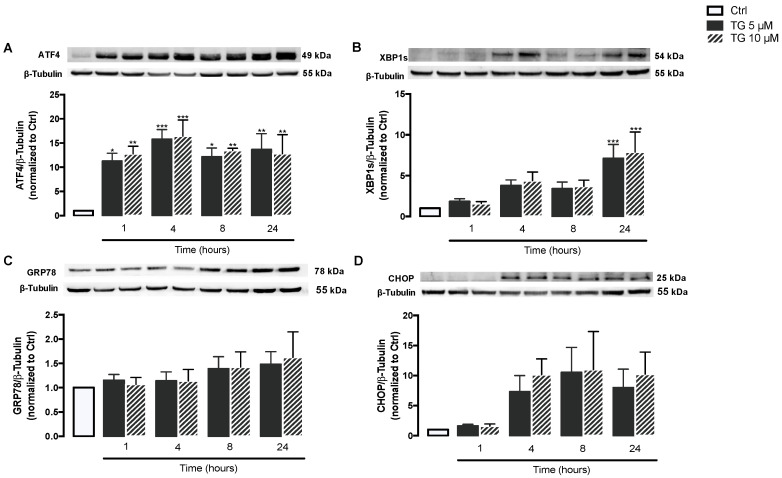
ER stress-induced UPR upon SERCA inhibition in TG-treated human THP-1 monocytes. Protein levels of ER stress markers, namely ATF4 (**A**), XBP1s (**B**), GRP78 (**C**), and CHOP (**D**), were quantified by WB in total cellular extracts obtained from human THP-1 monocytes treated with 5 or 10 μM thapsigargin (TG) during the indicated time periods (1–24 h). β-Tubulin I was used to control protein loading and to normalize the levels of the protein of interest. Results were calculated relatively to control values and represent the means ± SEM of at least three independent experiments. Statistical significance between control (untreated cells) and TG-treated cells was determined using the one-way ANOVA test, followed by Dunnett’s post hoc test: * *p* < 0.05; ** *p* < 0.01; *** *p* < 0.001.

**Figure 2 cells-11-00433-f002:**
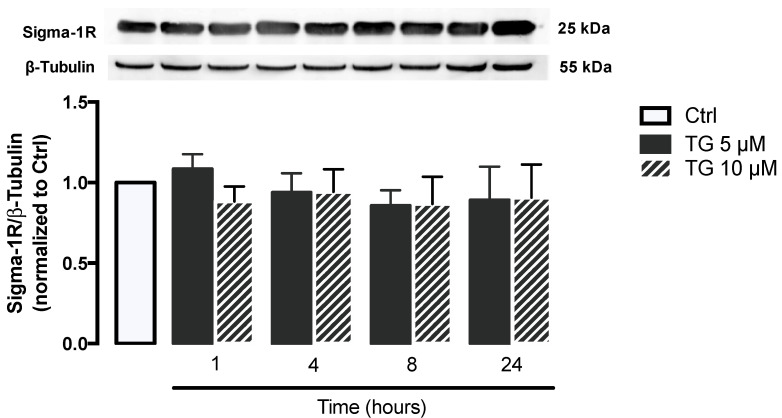
Stress response under stress conditions triggered by TG-induced SERCA inhibition in human THP-1 monocytes. Protein levels of Sigma-1R were quantified by WB in total cellular extracts obtained after treatment of THP-1 cells with 5 or 10 μM thapsigargin (TG) during the indicated time periods (1–24 h). β–Tubulin I was used as a control for protein loading and to normalize the levels of the protein of interest. Results were calculated relatively to control values and represent the means ± SEM of at least three independent experiments. Statistical significance between control and TG-treated cells was determined using the one-way ANOVA test, followed by Dunnett’s post hoc test.

**Figure 3 cells-11-00433-f003:**
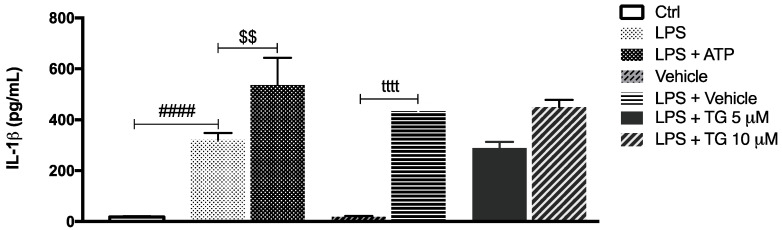
IL-1β secretion upon TG-mediated SERCA inhibition in THP-1 monocytes. Levels of secreted IL-1β were quantified by an ELISA assay in supernatants of THP-1 cells treated with 1 μg/mL LPS alone (24 h), with LPS (24 h), and then with 5 or 10 μM TG for the last 8 h. Cells primed with 1 μg/mL LPS for 24 h and then exposed to 5 μM ATP for 30 min were used as a positive control for NLRP3 activation. Results were expressed by pg/mL and represent the means ± SEM of at least three independent experiments. Statistical significance between LPS and control conditions (Ctrl), LPS and LPS plus ATP, and vehicle and vehicle plus LPS was determined by Student’s *t*-test (^####^
*p* < 0.0001, ^$$^
*p* < 0.01, ^tttt^
*p* < 0.0001); and between vehicle plus LPS and LPS plus TG-treated cells was determined using the one-way ANOVA test, followed by Dunnett’s post hoc test.

**Figure 4 cells-11-00433-f004:**
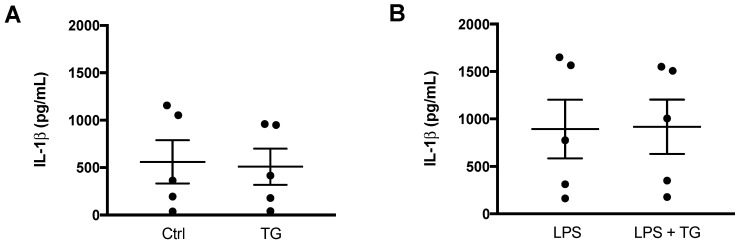
NLRP3 inflammasome activation in primary human monocytes. An ELISA kit was used to quantify the levels of IL-1β in supernatants of untreated (Ctrl) and TG-treated primary human monocytes during 8 h (**A**) and in supernatants of monocytes treated with 1 μg/mL LPS alone (24 h), or with LPS (24 h) and then with 10 μM TG for the last 8 h (**B**). Results represent the means ± SEM results obtained in samples from 5 participants. Statistical significance was determined by Student’s *t*-test.

**Figure 5 cells-11-00433-f005:**
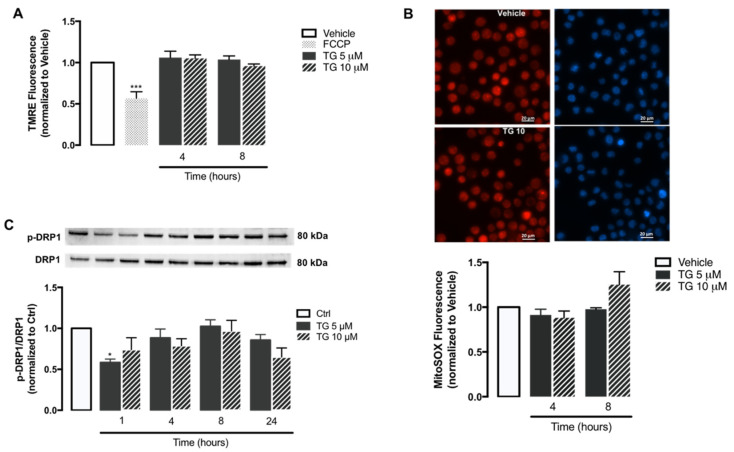
Mitochondrial membrane potential, ROS accumulation, and fission upon TG-induced SERCA inhibition in human THP-1 monocytes. Alterations in mitochondrial membrane potential (**A**) and mitochondrial ROS retention (**B**) were measured in THP-1 monocytes treated with 5 or 10 µM TG for 4 or 8, using the TMRE and MitoSOX fluorescent probes, respectively. FCCP was used as a positive experimental control for membrane depolarization. Results were calculated relatively to vehicle values and represent the means ± SEM of three independent experiments. Statistical significance between vehicle and TG-treated cells was determined using one-way ANOVA test, followed by Dunnett’s post hoc test: *** *p* < 0.001. p-DRP1 (Ser616) protein levels (**C**) were quantified by WB in total cellular extracts obtained after incubation of THP-1 cells with 5 or 10 μM TG during the indicated time periods (1–24 h). Total DRP1 was used to normalize p-DRP1 levels. Results were calculated relatively to control values and represent the means ± SEM of three independent experiments. Statistical significance between control and TG-treated cells was determined using the one-way ANOVA test, followed by Dunnett’s post hoc test: * *p* < 0.05.

**Figure 6 cells-11-00433-f006:**
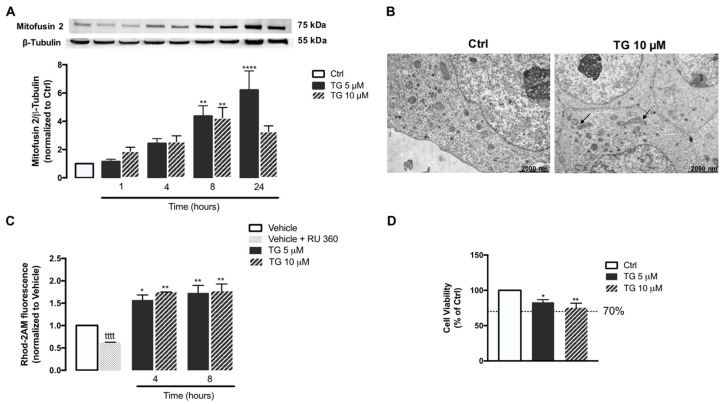
Mitochondrial fusion and Ca^2+^ content, and susceptibility towards ER stress under TG-induced SERCA inhibition in human THP-1 monocytes. Mfn2 protein levels (**A**) were quantified by WB in total cellular extracts obtained after incubation of THP-1 cells with 5 or 10 μM TG during the indicated time periods (1–24 h). β−Tubulin I was used as a control for protein loading and to normalize the levels of the protein of interest. Results were calculated relatively to control values and represent the means ± SEM of at least three independent experiments. (**B**) Representative TEM images of the mitochondrial morphology in TG-treated and untreated THP-1 monocytes. Arrows indicate fused mitochondria. ER-mitochondria Ca^2+^ transfer (**C**) in THP-1 cells treated with 5 or 10 µM TG for 4 or 8 h was detected with the fluorescent probe Rhod-2/AM. The selective inhibitor of MCU (RU 360, 10 μM) was used as a positive control for selective mitochondrial Rhod-2 uptake in THP-1 monocytes. Results were calculated relatively to vehicle values and represent the means ± SEM of at least three independent experiments. Susceptibility of THP-1 cells to 5 or 10 μM TG for 24 h was assessed by the resazurin assay (**D**). Results represent the mean ± SEM of at least three independent experiments and were calculated relatively to control values (untreated cells). Statistical significance between control or vehicle and TG-treated cells was determined using the one-way ANOVA test, followed by Dunnett’s post hoc test: * *p* < 0.05, ** *p* < 0.01, **** *p* < 0.0001. Student’s *t*-test was used to compare the statistical significance between vehicle and vehicle plus RU 360-treated cells (^tttt^
*p* < 0.0001).

## Data Availability

The datasets used and/or analyzed during the current study are available from the corresponding author on reasonable request.

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
