# Peer review of "Mitochondria Fusion upon SERCA Inhibition Prevents Activation of the NLRP3 Inflammasome in Human Monocytes"

_cells, 2022, doi:10.3390/cells11030433_

Round 1

Reviewer 1 Report

In this manuscript, Pereira et al reported the effects of SERCA inhibition by using TG in human cell line THP-1. This work shed light to better understand the molecular mechanisms underlying the Ca2+ signaling and ER stress-associated events in THP-1 cells. I have following comments:

  1. Please provide the full name for the abbreviations in the introduction such as ASC, NOD, DRP1.
  2. The discussion should be revised without repeating the results. Please remove the figures.
  3. The author should discuss the increase in mitochondrial Ca2+ levels after the treatment with TG; however, the mitochondrial membrane potential failed to change in THP-1 cells post the TG treatment at the same time points.
  4. Did TG treatment affect the cell number/proliferation of THP-1 cells?
  5. The author utilized a human monocyte cell line THP-1 for these studies, which may not similar and representative of freshly-isolated human monocytes from peripheral blood. Therefore, the conclusion should be careful. Only testing in THP-1 cells, the author could not claim the human innate immune system. Did the author test the effects of SERCA inhibition in freshly-isolated human blood monocytes?

Author Response

Reviewer 1:

  1. Please provide the full name for the abbreviations in the introduction such as ASC, NOD, DRP1.

R: As requested by the reviewer, the full name for the abbreviations in the introduction section were added. ASC is abbreviation for apoptosis-associated speck-like protein containing a CARD and it was added at line 89. Nod-like receptor pyrin domain containing 3 (NLRP3) was added at lines 87/88 and Dynamin-related protein 1 (DRP1) at line 100.

  1. The discussion should be revised without repeating the results. Please remove the figures.

R: As suggested by the reviewer, figures have been removed throughout the discussion section.

  1. The author should discuss the increase in mitochondrial Ca levels after the treatment

with TG; however, the mitochondrial membrane potential failed to change in THP-1 cells post the TG treatment at the same time points.

R: Based on our hypothesis, SERCA inhibition causes stress in monocytes, therefore, THP-1 monocytes treated with TG trigger an adaptive response that is mediated by the imbalance of mitochondrial dynamics towards mitochondrial fusion for enhanced mitochondrial Ca2+ uptake, in order to stimulate production and cell survival. Accordingly, recent studies demonstrated that modulating mitochondrial dynamics toward increased fusion markedly increases both mitochondrial Ca2+ retention capacity and Ca2+ uptake rates (https://doi.org/10.1096/fj.201901136R). Moreover, previous studies reported that an early upregulation of mitochondrial calcium occurs upon stress conditions in order to activate enzymes involved in the TCA cycle and, therefore stimulate ATP production (https://doi.org/10.1016/j.tcb.2008.12.002; https://doi.org/10.1016/j.ceca.2014.11.002). Also, ATP production is strictly dependent of mitochondrial potential and dissipation of transmembrane potential is associated with inhibition of mitochondrial fusion (https://doi.org/10.1134/S0006297915050053). In conclusion, data from literature supports that mitochondria fusion is associated with increased Ca2+ content and preservation of membrane potential, as demonstrated for THP-1 monocytes. In addition, the maintenance of mitochondrial polarization after TG treatment in these cells could also be associated with the activity of Ca2+extrusion systems that avoid mitochondrial Ca2+ overload (https://doi.org/10.3892/ijo.2019.4696), which causes loss of the mitochondrial membrane potential, ATP depletion and apoptosis. In the present study, only a slight decrease in the survival of THP-1 monocytes treated for 24 h with TG was observed (Fig. 6D), which further supports that mitochondria fusion (Fig. 6A, B), enhanced Ca2+ uptake (Fig. 6C) and maintenance of membrane potential (Fig. 5A) is a protective response against TG-induced stress.

  1. Did TG treatment affect the cell number/proliferation of THP-1 cells?

R: In this study we did not assess directly the proliferation of THP-1 cells, however, viability of TG-treated cells was evaluated by the resazurin assay, as indicated by Fig. 6D, which can reflect changes in proliferation. A decrease of cell viability by approximately 20 % was detected, which although reaching statistical significance is not considered biologically significant. Additionally, the analysis of MitoSOX (red) and Hoechst fluorescence (blue) fluorescence images obtained from both untreated and TG-treated cells, which were included in Fig. 5B, it can be observed that the number of cells is not different between experimental conditions. Therefore, it can be concluded that TG treatment does not affect the cell number/proliferation of THP-1 human monocytes.

  1. The author utilized a human monocyte cell line THP-1 for these studies, which may not similar and representative of freshly-isolated human monocytes from peripheral blood. Therefore, the conclusion should be careful. Only testing in THP-1 cells, the author could not claim the human innate immune system. Did the author test the effects of SERCA inhibition in freshly-isolated human blood monocytes?

R: We absolute agree with the comment of the reviewer and we understood the relevance of testing the effects of TG-induced SERCA inhibition in primary human monocytes. As intended by the reviewer, new graphs displaying the role of SERCA inhibition on human monocytes isolated from peripheral blood have been added to the revised version of the manuscript. The low yield of freshly-isolated human blood monocytes strongly impacted the number of assays performed with these cells. Due to this practical limitation, we carefully selected one experimental assay to carried out with these cells, having chosen to investigate the effect of SERCA inhibition on NLRP3 inflammasome activation (Fig. 4). For that, we measured the levels of IL-1b secreted by primary human monocytes treated with TG, in absence (Fig.4A) or presence of LPS priming (Fig. 4B). No differences were observed when comparing IL-1b levels secreted by TG-treated and non-treated monocytes indicating that, as previously observed in human THP-1 monocytes, SERCA inhibition doesn’t trigger NLRP3 inflammasome activation in primary human monocytes.

Taken together, our findings demonstrate that NLRP3 inflammasome is not activated upon SERCA inhibition in human monocytes, in opposition to what was previously in other innate immune cells, namely in macrophages.

Reviewer 2 Report

Ana Catarina Pereira et al. showed that human monocytes can counteract ER stress induced by SERCA inhibition by regulating mitochondrial morphology and preventing NLRP3 inflammasome activation. There are some recommendations:

  1. Why the time 4 or 8 h was chosen for the TG treatment in Figure 3-5 while ER stress was mostly induced at 24 h in Figure 1? TG was found to cause cell death in previous studies.
  2. There are three branches for ER stress. Thus it would be better to determine the protein expression for ATF6 branch in Figure 1.
  3. Please provide the representative fluorescence images in Figure 4 and 5.
  4. There are various cytokines. What is the rationale to only detect IL-1β but not the others?

Author Response

Reviewer 2:

  1. Why the time 4 or 8 h was chosen for the TG treatment in Figure 3-5 while ER stress was mostly induced at 24 h in Figure 1? TG was found to cause cell death in previous studies.

R: ER stress induced by TG in THP-1 human monocytes was shown to be an early event as demonstrated by the analysis of ATF4 levels, which increased at 1, 4 and 8 h (Fig.1A) that was followed by XBP1s upregulation (Fig.1B). Although the increase observed in XBP1s levels only reached statistical significance at 24 h, a tendency to increase at shorter time points, namely at 4 and 8 h, was observed. In addition, levels of Sigma-1R, a modulator of ER stress, were shown not affected upon TG treatment, at all tested time points (Fig.2), which might sustain ER stress. Altogether, these results contributed to choose 4 or 8 h for TG treatment as time points to continue the subsequent studies. Indeed, this work was aimed to investigate the role of ER stress occurring as a consequence of TG-induced SERCA inhibition on NLRP3 activation in human monocytes under early/moderate stress conditions, in order to understand whether adaptive mechanisms, and which ones, are involved in the response of human monocytes to stress. On the other hand, it was not our intention to address the role of chronic ER stress that can culminate in apoptotic cell death, which can be induced by TG as previously described. This is possibly also in the case of TG-treated monocytes, since increased levels of the pro-apoptotic factor CHOP were observed after 4 h of ER stress induction by TG (Fig. 1D), which can become statistically significant for longer incubation periods (e.g. 36, 48 or 72 h) or higher TG concentrations.

  1. There are three branches for ER stress. Thus it would be better to determine the protein expression for ATF6 branch in Figure 1.

R: We would like to thank the comment of the reviewer, which we understand, and we would like to clarify why we decided to focus on the PERK and IRE1a arms of the UPR. This study was aimed to investigate whether SERCA inhibition induced by TG, and subsequent induction of ER stress, activates NLRP3 inflammasome leading to sterile inflammation in human monocytes. Moreover, it also pretended to assess the relationship between mitochondrial dysfunction and NLRP3 inflammasome activation under these conditions. Therefore, the gold targets of this work were inflammation and mitochondria. The PERK and IRE1a UPR branches were the particular focus of this study, given their crucial role in triggering inflammatory responses during ER stress. Under these conditions, components of the PERK and IRE1a UPR arms interact with inflammatory signaling cascades, which results in the upregulation of inflammatory genes and subsequently in the induction of inflammatory responses. When activated by phosphorylation, the PERK downstream signaling mediator eIF2α is able to activate NF-κB by inhibiting the translation of Iκ. Activated IRE1 interacts with IKK and JNK, leading to the activation of the inflammatory mediators NF-κB and AP-1 (https://doi.org/10.14348/molcells.2018.0241). The IRE1a branch has a highly conserved immune function by mediating the activation of the NF-KB, and by modulating the development and survival of innate immune cell (https://doi.org/10.1038/ni.2991). In addition, PERK was also investigated because it was demonstrated that the mechanism that protects mitochondria during ER stress is orchestrated through the PERK branch of the ER UPR (https://doi.org/10.1016/j.celrep.2018.02.055).

  1. Please provide the representative fluorescence images in Figure 4 and 5.

R: First of all, we would like to thank and apologize in advance for not having included the representative fluorescence images. In the case of ROS analysis both MitoSOX and Hoechst 33342 images were included in the revised version of the manuscript at Fig. 5B, which corresponds to Fig. 4 in the previous version. Regarding Fig. 6, previously Fig. 5, related to the Rhod-2/AM assay, experiments were performed using a plate reader to evaluate calcium levels. We believe that the best approach to evaluate calcium levels should be a fluorescence microscope. However, the THP-1 cell line grows in suspension and it is not possible to adhere them in monolayer for microscopy experiments, which represents a serious practical limitation. To overcome that, we used a plate reader and added Ru 360 as a positive control to discard the possibility of probe mislocalization (Fig. 6C). 

  1. There are various cytokines. What is the rationale to only detect IL-1β but not the others?

R: I would like to thank the comment, which was taken in consideration. This work aims to investigate the role of SERCA inhibition on the activation of the NLRP3 inflammasome. According to the literature, when this multiprotein complex is activated, there is the cleavage of two pro-inflammatory cytokines leading to secretion of the mature isoforms: interleukin-1b (IL-1b) and interleukin-18 (IL-18). As largely reported in the literature (https://doi.org/10.1189/jlb.3A0716-300RR; 0.3390/ijms20133328) IL-1b is used as the classical readout of NLRP3 inflammasome activation. IL-1b is preferred over IL-18 because the activation of the inflammasome is normally associated with an inflammatory response, and IL-1bpreferentially involved in acute inflammatory responses, as demonstrated by the fact that IL-1b is able to activate the canonical NLRP3 pathway by activating NF-KB, which can lead to pyroptosis (inflammatory cell death).

Reviewer 3 Report

The manuscript of Pereira et al. describes the responses of a THP-1 monocytic cell line to the inhibitor of SERCA, a well-known ER stress inducer thapsigargin. Specifically, the authors questioned whether thapsigargin elicits NLRP3 inflammasome activation and alterations of mitochondrial function (mitochondrial membrane potential, ROS production, and changes of mitochondrial morphology) in THP-1 monocytes. Whereas thapsigargin induced ER stress, it failed to increase IL-1beta production in LPS-primed THP-1 cells. In addition, thapsigargin did not cause loss of mitochondrial membrane potential or elicited ROS production while inducing increased expression of mitofusin-2 protein favoring mitochondrial fusion.

As noticed by authors, the failure of thapsigargin to activate NLRP3 inflammasome was observed before in THP-1 macrophages (Ref 19). However, thapsigargin induced NLRP3 activation in murine bone marrow derived macrophages in that study. Furthermore, other ER stressors such as tunicamycin or brefeldin A induced NLRP3 activation in THP-1 macrophages, although in UPR effector-independent manner. Although the current study added some incremental information towards the reasons for the failure of thapsigargin to activate NLRP3 in THP-1 monocytes, it is completely unclear whether this effect is cell-type specific (e.g. does thapsigargin activate inflammasome in primary monocytes).

Most importantly, the choice of the cellular model is inappropriate to study NLRP3 activation, since THP-1 monocytes were reported to constitutively express active caspase-1 (e.g. PMID 24043885). In fact, data presented in the Figure 3 confirm that; LPS alone is able to elicit secretion of considerable amounts of IL-1beta, which is only slightly augmented by ATP.

Additionally, data provided by the authors on Figure 4 and 5 (levels of Drp-1 and Mfn2 proteins) do not allow to draw conclusions on the state of mitochondrial morphology in THP-1, which should be assessed directly (most preferably, electron microscopy).

Furthermore, measuring Rhod-2 fluorescence in a plate reader does not allow determination of the probe location. Since Rhod-2 is prone to mislocalization in the cytosol, the method used cannot be used to draw conclusions about mitochondrial calcium.

Specific comments:

Line 93 – fusion, not fission, is coordinated by mitofusins

Fig 4C – please state which phosphorylation site of Drp1 was assessed. The information on the antibody against phospho-Drp1 should also be added to methods section.

Author Response

Reviewer 3:

  1. As noticed by authors, the failure of thapsigargin to activate NLRP3 inflammasome was observed before in THP-1 macrophages (Ref 19). However, thapsigargin induced NLRP3 activation in murine bone marrow derived macrophages in that study. Furthermore, other ER stressors such as tunicamycin or brefeldin A induced NLRP3 activation in THP-1 macrophages, although in UPR effector-independent manner. Although the current study added some incremental information towards the reasons for the failure of thapsigargin to activate NLRP3 in THP-1 monocytes, it is completely unclear whether this effect is cell-type specific (e.g. does thapsigargin activate inflammasome in primary monocytes).

R: We would like to thank in advance the reviewer’s comment. In fact, we agree with the relevance of testing the effect of TG-induced SERCA inhibition in the activation of the NLRP3 inflammasome in primary human monocytes, in order to disclose whether this effect is cell-type specific. Therefore, taking advantage of our close collaboration with the University Hospital, we were able to isolate primary monocytes from peripheral blood collected from healthy subjects and investigate the role of SERCA inhibition on NLRP3 inflammasome activation in these cells. The obtained results are presented in Fig. 4 of the revised version of the manuscript. The levels of IL-1b secreted by primary monocytes treated with TG in the absence (Fig. 4A) or in the presence of LPS priming (Fig. 4B) were determined. No significant differences were observed in secreted IL-1b levels between TG-treated and untreated primary cells, as previously observed in THP-1 monocytes, demonstrating that indeed SERCA inhibition does not activates the NLRP3 inflammasome in human monocytes. Given that TG is able to promote NLRP3 inflammasome activation in bone marrow derived macrophages (Ref 19), but not in primary monocytes, it can be concluded that the ability of TG-induced SERCA inhibition to activate the NLRP3 inflammasome is a cell-type specific process, which can arise from the function played in the innate immune system. As described in the literature, macrophages are the main innate immune cells involved in acute inflammatory responses, which is not so exacerbated in their precursors, the monocytes (https://doi.org/10.5772/intechopen.88013).

  1. Most importantly, the choice of the cellular model is inappropriate to study NLRP3 activation, since THP-1 monocytes were reported to constitutively express active caspase-1 (e.g. PMID 24043885). In fact, data presented in the Figure 3 confirm that; LPS alone is able to elicit secretion of considerable amounts of IL-1beta, which is only slightly augmented by ATP.

R: Several studies reported in the literature evaluated NLRP3 inflammasome activation in THP-1 monocytes. For instance, the article mentioned by the reviewer (e.g. PMID 24043885) found that palmitic acid, the major dietary saturated fatty acid, induces NLRP3 inflammasome-mediated IL-1βproduction in this cell line. Based on the canonical pathway for NLRP3 inflammasome activation (DOI:10.1038/nm.3893), this is a 2-signal process that involves a priming signal (signal 1), which increases the expression of NLRP3 and pro-IL-1b, and a signal 2 that promotes the assembly of the three NLRP3 inflammasome components (NLRP3, ASC and pro-caspase-1) culminating caspase-1 activation and cleavage of pro-IL-1b into IL-1b .  Thus, in the absence of the priming and assembly steps there is no activation of the NLRP3 inflammasome and subsequent secretion of IL-1b,  even with a constitutive expression of active caspase-1. Recently, we observed that a different stimulus activated the NLRP3 inflammasome on THP-1 monocytes leading to a significant secretion of IL-1b , and results are presently under review in another journal, supporting the use of this cell line to investigate the activation of the NLRP3 inflammasome. LPS acts simultaneously as a priming and assembly signal for NLRP3 inflammasome activation (https://doi.org/10.3390/ijms20133328), therefore it is expected to find a significant increase of IL-1b levels secreted by cells exposed to LPS. Concerning the levels of IL-1bsecreted by monocytes treated with LPS alone or with LPS plus ATP, additional data is provided in the revised version of the manuscript (Fig. 3). By increasing the number of experiments, a great statistical difference between the two groups was observed.

  1. Additionally, data provided by the authors on Figure 4 and 5 (levels of Drp-1 and Mfn2

proteins) do not allow to draw conclusions on the state of mitochondrial morphology in THP- 1, which should be assessed directly (most preferably, electron microscopy).

R: We would like to thank the reviewer’s comment, which we considered that can significantly improve the scientific quality of the work. As suggested, representative images from transmission electron microscopy (TEM) obtained in untreated and TG-treated THP-1 monocytes were added to the revised version of the manuscript (Fig. 6B). TEM images show that the presence of fused mitochondria is more frequent in monocytes treated with 10 µM TG when compared to untreated THP-1 cells, further supporting the conclusions made based on the analysis of protein levels of the fusion protein Mfn2 in these cells. Taken together, our data suggest that TG treatment affects mitochondrial morphology, by increasing mitochondrial fusion. 

  1. Furthermore, measuring Rhod-2 fluorescence in a plate reader does not allow determination of the probe location. Since Rhod-2 is prone to mislocalization in the cytosol, the method used cannot be used to draw conclusions about mitochondrial calcium.

R: We understand the point of view of the reviewer and we agree that the best approach to evaluate calcium levels shouldn’t use a plate reader but a fluorescence microscope. However, the THP-1 cell line grows in suspension and it is not possible to adhere them in monolayer for microscopy experiments, which represents a serious practical limitation. In order to assess the specificity of the Rhod-2AM fluorescent probe to measure mitochondrial calcium, overcome its mislocalization to the cytosol, untreated THP-1 monocytes (control cells) were incubated with the selective inhibitor of the Mitochondrial Calcium Uniport (MCU), Ru 360. By incubating cells with Ru 360, mitochondrial calcium uptake is blocked and therefore, a significant reduction of Rhod-2 fluorescence should be observed in the presence of Ru 360 if Rhod-2 fluorescence results from its accumulation in the mitochondria. Since Rhod-2 fluorescence significantly decreased in THP-1 cells treated with Ru 360 in comparison with untreated cells, it means that Rhod-2 is accumulated in mitochondria and changes in fluorescence arise from changes in Ca2+ levels within this organelle. Results were included in Fig. 6C on the revised version of the manuscript.

Specific comments:

Line 93 – fusion, not fission, is coordinated by mitofusins.

R: I would like to apologize in advance for this mistake and also to thank for the adequate correction. As kindly indicated by the reviewer, in the introduction section at line 99, the sentence “Mitochondrial fission is coordinated by mitofusins (MFN1, MFN2)” was replaced by Mitochondrial fusion is coordinated by mitofusins (MFN1, MFN2)”.

Fig 4C – please state which phosphorylation site of Drp1 was assessed. The information on the antibody against phospho-Drp1 should also be added to methods section.

R: As required by the reviewer, the phosphorylation site of Drp1 detected by the antibody against phospho-Drp1 was added to the material and methods sections, as well as to the results section and the legend of Fig. 5. In both cases, it was clearly indicated that the phosphorylation site of the antibody phospho-Drp1 is at Serine 616 (Ser 616).

Round 2

Reviewer 2 Report

The authors addressed my suggestions and I have no further comments.

Reviewer 3 Report

The authors satisfactory addressed my critique points.